# Exploring Irrigation and Water Supply Technologies for Smallholder Farmers in the Mediterranean Region

Dina Pereira [1,2], Joao Carlos Correia Leitao [3], Pedro Dinis Gaspar [4], Cristina Fael [5], Isabel Falorca [5], Wael Khairy [6], Nadya Wahid [7], Hicham El Yousfi [8], Bassou Bouazzama [9], Jan Siering [10], Harald Hansmann [11], Jelena Zascerinska [10,*], Sarah Camilleri [12], Francesca Busuttil [12], Malcolm Borg [12], Joseph Mizzi [13], Reno Micallef [13] and Joseph Cutajar [13]

1. Centre for Management Studies of Instituto Superior Técnico (CEG-IST), University of Lisbon, 1049-001 Lisbon, Portugal
2. NECE–Research Center in Business Sciences, Universidade da Beira Interior, 6201-001 Covilhã, Portugal
3. Faculty of Social and Human Sciences, NECE–Research Center in Business Sciences, Universidade da Beira Interior, 6201-001 Covilhã, Portugal
4. Department of Electromechanical Engineering, Universidade da Beira Interior, 6201-001 Covilhã, Portugal
5. Department of Civil Engineering and Architecture, Universidade da Beira Interior, 6201-001 Covilhã, Portugal
6. Civil Engineering Department, Heliopolis University for Sustainable Development, Cairo 11785, Egypt
7. Faculty of Science and Technology, Sultan Moulay Slimane University, Beni-Mellal 23000, Morocco
8. Faculty of Legal, Economic and Social Sciences Souissi, Université Mohammed V de Rabat, Rabat 10000, Morocco
9. National Institute of Research Agronomics of Morocco, Rabat 10090, Morocco
10. Hochschule Wismar, 23966 Wismar, Germany
11. Institut für Polymer- und Produktionstechnologien e. V., 23966 Wismar, Germany
12. The Malta College of Arts, Science and Technology, QRM 9075 Paola, Malta
13. EcoGozo Directorate, Victoria, VCT 1335 Gozo, Malta
* Correspondence: jelena.zascerinska@hs-wismar.de; Tel.: +49-1515-1467577

**Abstract:** Water security is a hot topic all over the world, due to global warming, climate change, natural calamities such as droughts and floods, overuse of water, and other factors. Water issues have been scientifically investigated from several perspectives, namely institutional, economic, social, environmental, managerial, and technological. However, the technological aspects of irrigation and water supply for smallholder farmers in the Mediterranean region have not been adequately addressed. This paper explores irrigation and water supply technologies for smallholder farmers in the selected Mediterranean countries (Egypt, Malta, Morocco, and Portugal). The methods of analysis are literature review, fieldwork, and observation. The literature survey reveals that Mediterranean countries share many common features in terms of climate, water and land resources, and development issues. Nevertheless, the selected countries in the Mediterranean region (Egypt, Malta, Morocco, and Portugal) differ in terms of type of crops, water management regulations, labor force availability, financial sustainability, and economic approaches. These remarks signal the need for applying a specific approach in selecting a technology for irrigation and water supply according to the regional context. Additionally, the financial and economic perspectives of the three key technologies (i.e., SLECI, desalination technology, and engineering constructed wetlands) require further analysis.

**Keywords:** agriculture; water-saving irrigation technologies; desalination technology; engineering constructed wetland; irrigation efficiency; Mediterranean region; self-regulating low energy claytube irrigation (SLECI); smallholder farmers; sustainable development; water scarcity; water supply

## 1. Introduction

Water scarcity is a critical issue for the agricultural sector in the Mediterranean region. The implementation of smart technology transfer mechanisms is expected to help, on the one hand, to increase the efficiency of irrigation systems and, on the other hand, to

anticipate the effects of climate change, namely prolonged periods of extreme drought and consequent water rationing [1]. There are 24 Mediterranean countries in total; this paper focuses on Egypt, Malta, Morocco, and Portugal. The selection of these four countries is based on their participation in the MED-WET project "Improving MEDiterranean irrigation and Water supply for smallholder farmers by providing Efficient, low-cost and nature-based Technologies and practices", supported by the Partnership for Research and Innovation in the Mediterranean Area (PRIMA) program [2]. This three-year project started in November 2021 and is currently in the process of installing the irrigation technologies in pilot sites in Malta, Morocco, and Portugal.

The MED-WET project was developed to ultimately improve the irrigation efficiency of smallholder farmers in the Mediterranean region, particularly through the optimal use of scarce water resources for lasting food and water security [2]. This will be done, amongst other methods, by:

1.　Developing new irrigation technologies and solutions widely applicable for small-holder farmers;
2.　Equipping smallholder farmers with knowledge and skills to install, adapt and operate more efficient and effective irrigation options;
3.　Increasing irrigation water availability from salinized and secondary sources;
4.　Enhancing farm profitability and environmental footprint of pilot farming practices.

For this to be achieved, three major technologies will be utilized in the project. The "SLECI" (Self-regulating, Low Energy, Clay based Irrigation) technology is a self-regulating subsurface irrigation technique that uses the actual suction force of the surrounding soil for regulation of the system's water release via clay tubes. Its concept, production and installation are simple, and thus adaptable to rural environments saving on water and energy. The project will investigate its performance in a variety of local conditions and in combination with various (reclaimed) irrigation waters. The second technology is a simple desalination system that will be used on saline and low-grade water to recuperate freshwater suitable for crop irrigation. Lastly, productive constructed wetlands will be used for wastewater reuse and its transformation into reclaimed irrigation water.

During the last decades, and in the face of climate change and environmental variability, agriculture has been forced to implement measures to increase efficiency and productivity, at the expense of resilience. Indeed, agricultural intensification (associated with intensive monoculture) has caused major impacts on environmental sustainability, such as landscape homogenization, soil contamination, and biodiversity reduction, through the widespread use of fertilizers, phytopharmaceuticals, and other resources [1]. Currently, it is common to find significant loss of production and/or increased costs due to the loss of biodiversity. In addition to ensuring food security, the sector employs a considerable share of the population and contributes significantly to the economy [1]. Agriculture is facing a shrinking workforce, as well as increased consumer demand for more transparent, sustainable, environmentally friendly, and high-quality products [3]. The new Common Agricultural Policy, in line with the European Ecological Pact, is focused on environmental protection, and will provide more incentives for bio-farmers who use the technique of cultivation without use of chemicals. Indeed, biodiversity plays a critical role in sustaining the natural cycles essential to agricultural ecosystems, as well as in supporting the conservation of ecological balance. It also promotes the emergence of new social and economic values, such as the creation of organic products, nature tourism, and the vitality of society [3]. Although the climatic conditions in the Mediterranean area are suitable for growing a wide variety of crops, irrigation is essential to maintaining consistent yields [3]. Thus, there is an urgent need to change the production paradigm to minimize environmental impact and promote local biodiversity without compromising productivity, with water availability playing a major role in reaching this objective [3].

Water issues in farming in the Mediterranean region have been addressed from different perspectives. For concerted water governance, the social-ecological Systems (SES) framework was developed [4]. Irrigation policies in the Mediterranean were revised [5].

This research analyzed changing relationships between irrigation and the wider water sector based on supply- and demand-management options, recent technical and institutional changes in the irrigation sector before turning to the importance of economic dimensions, and context [5]. Adaptations in irrigated agriculture in the Mediterranean region revealed that both biophysical and socioeconomic factors determine the context in which adaptations are implemented, and that considerable spatial variability in the area exists [1].

Water security is becoming a hot topic all over the world. Global warming, climate change, natural calamities such as droughts and floods, overuse of water, and other factors, exacerbate the situation with irrigation and water supply for smallholder farmers in the Mediterranean region. In Figure 1, seasonal and annual climate evolution in the Mediterranean region between 1950 and 2022 is demonstrated. Colors show the magnitude of changes in precipitation (left) and potential evapotranspiration (right), in mm. Black isolines represent areas with significant trends ($p < 0.05$) [6].

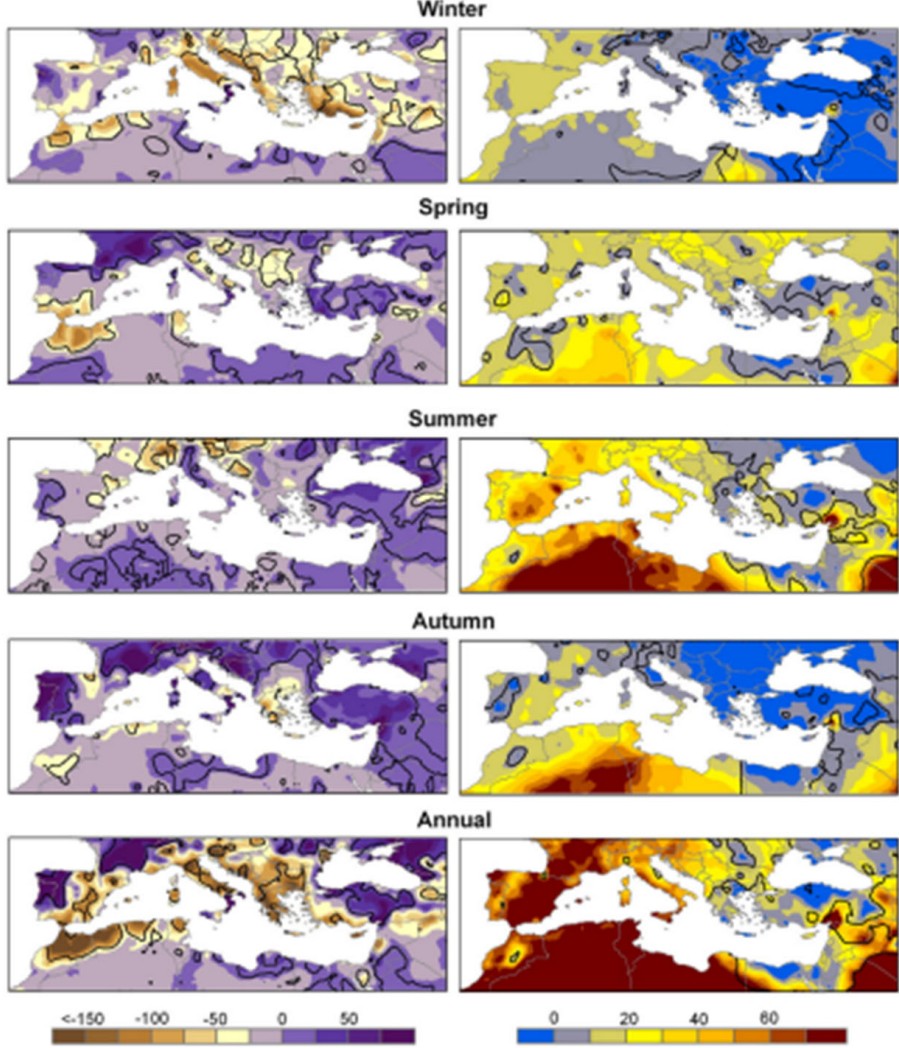

**Figure 1.** Seasonal and annual climate evolution in the Mediterranean region between 1950 and 2022. Colors show the magnitude of changes in precipitation (**left**) and potential evapotranspiration (**right**), in mm. Black isolines: areas with significant trends ($p < 0.05$) [6].

Meteorological conditions translate into medium-to-high values of the RDrI-Agri (Risk of Drought Impact for Agriculture) in a wide region along the coast of northern Africa, including Morocco [7]. RDrI-Agri ranges from low to medium values in central Portugal as shown in Figure 2 [7].

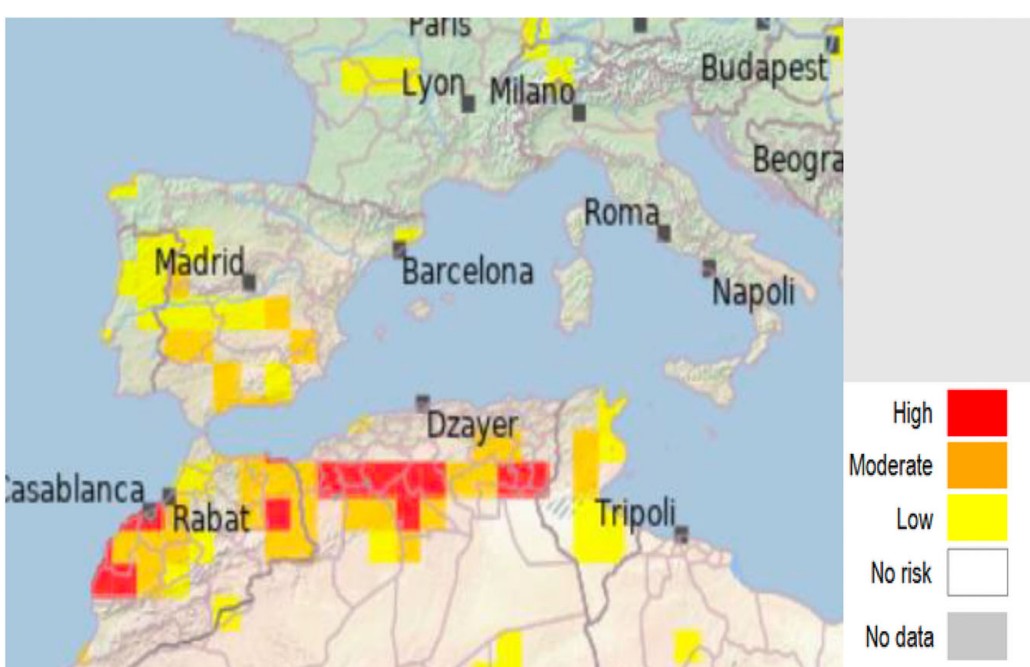

**Figure 2.** Risk of Drought Impact for Agriculture (RDrI-Agri)–first ten days of February 2022. Reprinted from [7].

Drought vulnerability in northern Africa is significantly higher, due to their lower GDP and a larger part of their society being dependent on rain-fed agriculture [4].

In the past decade, the policy of economical use of irrigation water has become a central issue in many irrigated regions of the Mediterranean [5,8], which face aging irrigation infrastructure, shortage of water needed for irrigation, climate change (drought), and population growth impacting food security. Thus, the Mediterranean strategy for sustainable development invests in the management of water demand through the improvement of agronomic and economic efficiency [4,9,10]. This encourages better use of existing water supplies aimed at increasing water saving and benefiting from a new technique for mobilizing water in irrigation.

The analysis of previous research confirms that a variety of water issues such as irrigation, water governance and irrigated agriculture was discussed. The water issues were investigated from a holistic point of view, although the technological aspects of irrigation and water supply for smallholder farmers in the Mediterranean region still remain under-explored.

The approaches to solving irrigation and water supply problems should be brought in line with the current situation in the countries concerned. This must be considered from economic and environmental perspectives, which are directly related to climate change, pollution, and other related environmental aspects, i.e., new technologies targeted to cleaner production and sustainable development.

The current international research program, discussed in this paper, is intended to provide all stakeholders involved in irrigation and water supply initiatives in the Mediterranean region with a sound foundation for analysis of the conditions and resources available for smallholder farmers. The research study is being implemented from 2021 to 2024.

The purpose of the paper is to explore irrigation and water supply for smallholder farmers in the four Mediterranean countries represented in the international project MED-WET. Our initial findings refer to the use of key technologies, namely SLECI, desalination technology, and engineering constructed wetland, for irrigation and water supply for smallholder farmers in the Mediterranean region. Our initial findings are based on the analysis of nature conditions and natural resources in the selected Mediterranean countries (Egypt, Malta, Morocco, and Portugal).



## 2. Materials and Methods

Literature review was selected, as it serves as the ground for future research and theory [11]. A narrative literature review [12] is performed in this work, allowing for assessment of multiple perspectives (e.g., researchers, smallholder farmers, entrepreneurs, businessmen, policy makers, governmental representatives, etc.), where the studies were carried out, and what study gaps there seem to be.

The narrative analysis draws a detailed and holistic picture of the analysis of nature conditions and natural resources in the selected Mediterranean countries that impact irrigation and water supply. The review establishes what technologies support irrigation and water supply for smallholder farmers in Egypt, Malta, Morocco, and Portugal.

As a method of data collection, hybrid (conventional and online) fieldwork is to be implemented [13]. According to Baczko and Dorronsoro [14], the practice of fieldwork tends to empower researchers, lets them produce their own hypotheses, and puts distance between themselves and the sorts of Taylorism making a strong comeback. It should be pointed out that Taylorism refers to the management science focused on the improvement of an organization's economic efficiency through the analysis of relationships between engineering processes and labor productivity. Fieldwork implies the collection of data in a real environment, compared to a survey, allowing for the observation of phenomena in their real-life conditions and cultural circumstances [14]. Observation is a highly effective method of obtaining qualitative data, utilizing several techniques, namely respondent interviews and self-analysis [15,16]. Moreover, observation contributes to a more adequate picture that emerges of the research setting as a social system described from several participants' perspectives [17].

The current study uses a conceptual framework [18] combining soil type, crop types and other factors.

### 2.1. Soil Type

In natural conditions, soil type can be identified as the most important factor because some soils (essentially sandy ones) have a higher infiltration rate, so they need more regular, but smaller, amounts of water. However, in the case that there is a variety of soils within a single irrigation scheme, sprinkler or drip irrigation are recommended. We then have the slope condition, which would influence the selection of sprinkler or drip irrigation. Then we must consider the climate condition because strong winds, for example, can disturb the spray of water sprinklers.

### 2.2. Crop Types

When considering the type of crop factor, sprinkler and drip irrigation are mostly used for high-value crops, such as vegetables and fruit trees, due to their high investment per hectare. Drip irrigation is better suited to single trees or row crops.

### 2.3. Farmer Experience

The previous experience with irrigation must also be considered, because an inexperienced farmer or producer using irrigation technologies can lead to unexpected complications in the process, and then the costs would be higher than the benefits.

The required labor inputs must be considered, since technologies always require some level of labor cost for installation, operation, and maintenance. For example, high-performance surface irrigation requires accurate land leveling, and a high level of farmer's organization to operate the system as effectively as possible. Drip and sprinkler irrigation also requires regular maintenance to perform adequately.

Then, the cost and benefits of the process must be considered. Before choosing an irrigation method, the costs must be estimated for all the viable options. The construction, installation, and operation and maintenance costs must be considered when estimating the expected benefits of the of yield increasing from technical changes.

Materials available for our investigation were a determining factor in the implementation of methods of analysis.

Mediterranean countries share many common features in terms of climate, water, and land resources, as well as development issues [19]. These include arid and semi-arid climate, limited water resources, agricultural development limited by water availability, and high economic and social value of water [19].

## 3. Results

This section gives an overview of the results obtained within the investigation carried out in each selected country. Table 1 presents the summary of the four countries' characteristics.

**Table 1.** The summary of the four countries' characteristics.

| Characteristics | Egypt | Malta | Morocco | Portugal |
|---|---|---|---|---|
| Nature conditions | Desert | Semi-arid and calcareous, sandy, loamy, clay soil | Semi-arid and silty-clay soil | Arid, acidic, and sandy, rocky soil |
| Historical evolution of irrigation and water supply | From the fixed fresh surface waters to virtual water imports | Rainfall, groundwater, reverse osmosis, new water | From well water through dam water to saved water | From a set of hydroelectric concessions to integrated management of water resources and ecosystems |
| Main irrigation technologies | Engineering constructed wetland | Drip-irrigation technologies | Superficial drip irrigation system | Conventional drip system, including nano-irrigation |

### 3.1. Egypt

3.1.1. Nature Conditions

The primary source of irrigation in Egypt is the river Nile. The maximum quantity of fresh water that Egypt can use from the Nile River is fixed at 55.5 billion $m^3$ annually. However, with increasing demand, dependency on groundwater has grown over the years. In 2018, non-renewable deep groundwater resources and shallow groundwater abstraction in the Nile Valley and Delta were 2.1 and 7.5 billion $m^3$/year, respectively. Shallow groundwater is found in the Nile Delta aquifer, a semi-confined aquifer that is at high risk of salinization, and yet pumping is uncontrolled. The aquifer is recharged by infiltration from excess irrigation estimated between 0.25 and 0.8 mm/day, and by seepage from the irrigation canal system, while some other canals are recharged by groundwater leakage. On the other hand, deep groundwater is found in western and eastern deserts as well as in Sinai, at depths of over 600 m, with reasonable quality for human use [20].

In Egypt, due to the aridity and limited freshwater resources, the total cultivated area mainly using surface irrigation is about 3.5% of the total area of the country. The overall water use efficiency in Egypt reaches over 75%. The Nile Delta contains about 65% of irrigated land in Egypt [21]. The agricultural sector is the main consumer of surface water and groundwater (GW). Surface water being used for irrigated agriculture accounts for about 85% of the total surface water available, while GW for agriculture accounts for about 80% of total groundwater withdrawals. The escalating rate of GW abstraction resulted in seawater intrusion and soil salinization of some northern parts of the Nile Delta. In addition to over-abstraction, sea level rise, temperature rise due to climate change, soil formations, and changes in the flow of the Nile, also contribute to the salinization of the aquifer [22].

The situation in the deserts of Egypt is different. The Nubian Sandstone Aquifer System (NSAS) is one of the largest aquifer systems in the world, with an area of more than 2.0 million $km^2$ crossing the borders, from largest to smallest spatial extent, of Egypt, Libya, the Sudan, and Chad.

The study area under the MED-WET project (engineering constructed wetland pilot for irrigation, planned to be developed, adopted by, and benefiting about 25,000 smallholder farmers) is located in Sekem, as shown in Figure 3, for land Reclamation farms in El-Wahat El-Baharia, Egypt, which is part of the Western desert within the domain of NSAS. So,

the main source of irrigation and domestic water in the study area comes from the upper
NSAS aquifer.

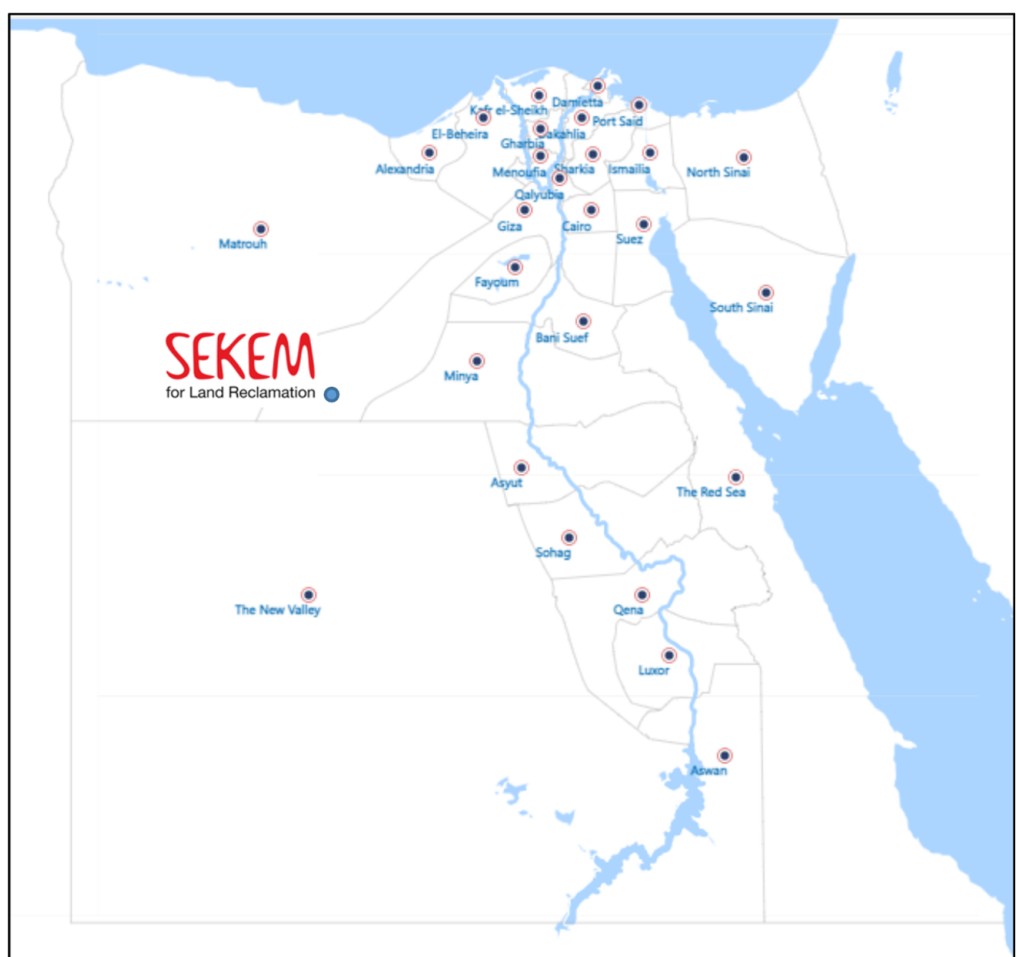

**Figure 3.** Map of Egypt showing the study area of engineering constructed wetland at Sekem for
land Reclamation in El-Wahat El-Baharia, Giza [23].

3.1.2. Historical Evolution of Irrigation and Water Supply in Egypt

The agriculture sector in Egypt uses about 63.0 billion m$^3$ of water annually. About
40.0 billion m$^3$/year are withdrawn for agriculture purposes from the fixed fresh surface
waters of the Nile River and its irrigation network. In addition, the coastal rainfall is in the
amount of about 1.3 billion m$^3$/year. About 7.5 billion m$^3$/year are abstracted from excess
agricultural return to the shallow groundwater within the Nile Valley and Delta. About
9.3 billion m$^3$/year of agricultural drainage water is being reused in agriculture, in addition
to about 4.2 billion m$^3$/year of treated wastewater being directly or indirectly reused in
agriculture. The remaining amount necessary for agriculture purposes is compensated
from withdrawals from non-renewable groundwater aquifers. Most of the Nile waters in
Egypt are used and recycled several times, to balance the water needs with the available
water resources.

The role of virtual water imports in achieving Egypt's food security is fundamental. It
reflects on existing water policies and provides actions for those policies that would achieve
quick wins in the future. So far, Egypt imports virtual water (strategic crops and meats)
equivalent to the amount of 34 billion m$^3$ of water every year [20]. The impacts of climate
change on the water sector in Egypt are severe, due not only to increasing population and
escalating temperature that boost water scarcity and evaporation, but also due to about

97% of its water resources originating from other upstream Nile Riparian countries which aim at exploiting more Nile water for its local sustainable development.

### 3.1.3. Main Irrigation Technologies in Egypt

With the surface water resources of Egypt currently fully exploited, and the groundwater pumping reaching the maximum limit, the need for alternative water resources has never been of more profound urgency than it is nowadays. Wastewater treatment is listed at the top of those alternatives. Due to the technology involved, the cost of wastewater reuse exceeds that of potable water in Egypt, like other regions of the world, especially where a fresh water supply is conveniently available. However, service of treated wastewater is usually subsided to citizens at below cost, in order to encourage its use. In 2014 it was recorded that Egypt produced about 7.0 billion $m^3$/year of wastewater [24], about 3.7 billion $m^3$/year of which were untreated, 2.4 billion $m^3$/year of which were secondarily treated, 0.9 billion $m^3$/year of which were primarily treated, and only about 0.1 billion $m^3$/year of which were tertiarily treated. Out of that total (about 3.4 billion $m^3$/year) treated wastewater, only about 0.3 billion $m^3$/year were reused directly for agriculture, while the remaining amount was disposed of to the national drainage network where they were indirectly reused [25]. In 2017, according to HCWW (Holding Company for Water Supply and Wastewater), about 96.6% of the collected wastewater was safely treated. The Egyptian Code of practice for the reuse of treated wastewater for agricultural purposes, ref. [26] issued by the Ministry of Housing, Utilities and Urban Communities of Egypt, placed strict categorized safeguards and restrictions for the healthy and sustainable uses of each grade (A, B, C, and D) of the treated wastewater [27].

Engineering constructed wetland treatment (CWT) as depicted in Figure 4 fits well for the rural and desert communities because it is a low-cost (does not need energy nor sophisticated technicians), natural-based, and efficient wastewater treatment technology.

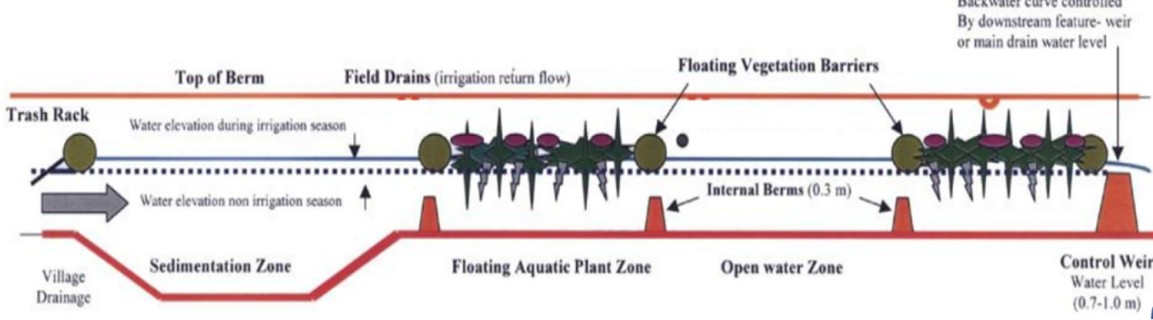

**Figure 4.** Conceptual Design of the engineering constructed wetland treatment pilot.

Also, it can be readily constructed in remote areas. CWT is efficient in treating municipal effluents, agricultural drainage, and animal wastes, which are the most common in such remote communities. CWT uses native special weeds, substrate, microorganisms and aeriation weirs, to remove contaminants from wastewater by mimicking the processes in natural wetland ecosystems with high efficiency.

### *3.2. Morocco*

#### 3.2.1. Nature Conditions

Morocco is one of the countries most threatened by water scarcity [28]. It is ranked 22nd among the countries most exposed to the risks of water insecurity. Water resources in Morocco are under different pressures, namely the increase in water demand due to the demographic explosion, recurrent droughts, irregular rain-fall because of climate change, and finally the requirements of economic development [29].

Faced with the scarcity of water, the Moroccan State has gradually improved the legislative mechanism governing the water sector (surface and underground), and in

adapting to new constraints, whether they be climatic, technological, and/or economic [28]. For surface water, Morocco has invested heavily in building dams and infrastructure to meet the country's water needs, particularly those of irrigated agriculture [28]. The precipitation received upstream of these basins totals about 19.3 billion m$^3$, of which only 17.7 billion m$^3$ can be mobilized [28]. Namely, 72% of the mobilizable waters are characteristic of the Nordic, Riffian and central hydraulic dams of the country. The hydraulic dams of the south contribute only 5% of the mobilizable water [28]. For groundwater, the exploitable potential of groundwater is estimated at 4 billion m$^3$/year.

The development of irrigation has led to the introduction of new production techniques and technologies aimed at improving agricultural productivity. The adoption of technological advances in irrigation has allowed for the considerable intensification of agricultural production. However, because of the succession of drought years during 2010–2020, the hydraulic dams show a water deficit of nearly 45% compared to a year of normal precipitation. In view of this water stress, the State decided from 1 October 2023, to stop irrigation by the dams of the center (Provinces of Tadla and El Haouz) and the south (Province of Souss-Massa and Darâa-Tafilalt). In this context of scarcity, the development of technological keys to water control and management for efficient and sustainable irrigated agriculture is desirable.

### 3.2.2. Historical Evolution of Irrigation and Water Supply in Morocco

Since 1956, when Morocco officially gained independence, irrigated agriculture remains a priority sector for government efforts. The irrigated land occupies only 15% of the cultivated area of the national acreages [30]. It contributes to a total value of 45% on average of the agricultural added value, and supports 75% of agricultural exports [31]. However, before 2007, despite the important contribution of the agricultural sector to the national GDP, its share in the budget of public investment remained limited [32]. This deficit of public investment, associated with a deficit of private investment, partly due to limited participation of the banking structure to finance the agricultural sector, had placed agriculture in a spiral of low productivity, low incomes, and poverty. In addition, for a long time, the sector was not properly diversified, with cereal production being the most prevalent [33]. Furthermore, the agricultural sector experienced strong growth volatility that negatively impacted the growth of the whole economy. The sector was also weakened by the lack of human resources [34]. Agricultural professional organizations were poorly organized and structured. Another identified problem was that these structures could not meet the new requirements of modern agricultural problems [32]. In addition to these structural deficiencies, the environment sector can be characterized by the climatic change that threatens the durability of the systems of production, in particular the modes of sustainable use of water; the instability and volatility of the world markets of basic commodities that resulted in the rise of increasing prices, for example, food crises in 2007–2008; and changes in the global competing landscape, which lead to increasing requirements to the product quality traceability by the customers.

In 2008, Moroccan agriculture had to choose a direction between stagnation and a potential to mobilize and rehabilitate to solve the faced challenges. In this context, the Department of Agriculture gave an impetus by formulating the strategy of the Green Morocco Plan (PMV) in 2008 [35]. This strategy focused on the objective to make agriculture an engine of economic and social development by the transformation of the agricultural sector into a modern, competitive, and inclusive sector. The programs of subsidy and reorganization through the specialized structures of the Regional Offices of Agricultural Development (ORMVA) were initiated by the State. They supported the reorientation of agriculture towards new techniques and farming methods, the promotion of the improved and adapted crop varieties, and the development of irrigation techniques to save water.

To balance the supply of water resources and demand of irrigated agriculture, the Green Morocco Plan adopted the policy of control and management of the water articulated around four structuring programs:

1.  National Program of Saving Irrigation Water (NPSIW);
2.  Program of Extension of Irrigation (PEI);
3.  Program of the Rehabilitation and Safeguard of the Perimeters of Small and Average Hydraulics (PMH);
4.  Program of Promotion of the Public-Private Partnership (PPP).

3.2.3. Main Irrigation Technologies

Several types of irrigation techniques have been used in Morocco, including gravity irrigation, sprinklers, and localized drip irrigation.

The gravity irrigation technique is used by small farmers. It uses an open channel that brings water by gravity to smaller channels, irrigating the cultivated plots. This irrigation system uses a lot of water, especially since a lot of it is lost through evaporation.

The sprinkler technique is practiced for large crops (cereals, legumes, etc.). This irrigation is carried out as part of an integrated or underground irrigation system. Indeed, the water circulates in underground pipes, and it goes out to mobile pipes which distribute it to the crops via sprinkler systems. Water losses mainly by evaporation are linked to this type of irrigation.

The localized drip irrigation technique is very economical in water use since it only consumes what the plant needs. The water mobilized in the plot is brought directly to the foot of the plant.

Micro-irrigation or localized irrigation is Morocco's preferred technique for sustainable agriculture, adapting to climate change, and saving water. "Crédit Agricole du Maroc" has created a loan consisting of 80% pre-financing by a grant from the Agricultural Development Fund, and 20% depreciable loan. It thus encourages investment in economical localized irrigation techniques with 100% coverage of investments for areas of less than 5 ha. A large drop-by-drop reconversion program has been launched as part of the Green Morocco plan [28]. This program has contributed to the collective conversion of 57,500 Ha to drip irrigation. In 2022, 32,600 farmers have benefited from this drop-by-drop conversion. This is a possible solution to fill the significant rain deficit that impacts all agricultural sectors in Morocco.

For a blue water economy, Morocco is investing in more effective, efficient, and economical technological keys. For example, many desalination projects are underway. In the context of water scarcity and economy, Morocco is studying an underground localized irrigation system (SLECI), as part of the MED-WET project.

In Morocco, as part of this project based mainly on the evaluation of the performance of the SLECI technology to improve irrigation efficiency for small farmers, the commitment of small farmers was sought in the first place. For this reason, the Morocco team from INRA and USMS adopts the participatory approach by involving officials from the National Agricultural Advisory Office (ONCA), and farmers. After a series of visits and discussions conducted by INRA team, we obtained a commitment of two small farmers and two INRA researchers in two different ecological regions, namely Beni Mellal-Khénifra region and Daraa-Tafilalt region (Hydraulic basin of Tadla) and Errachidia region (Hydraulic basin Daraa-Tafilalt), as depicted in Figure 5.

In the Beni Mellal-Khénifra region there are three chosen sites distributed in the provinces of Beni Mellal (INRA 1), Fqih Ben Saleh (Farmer 1 "Riad field"), and Aataouia-Azilal (Farmer 2 "Ait Naceur Field"). For the Daraa-Tafilalt region and more specifically in the province of Errachidia, there are the two INRA sites.

Today's irrigation technologies can be divided into surface and subsurface irrigation. In addition to well-known surface irrigation methods (e.g., basin irrigation), sprinkler systems (e.g., spraying systems) characterize the picture of above-ground irrigation on cultivated land. To reduce evaporation losses at the soil surface, subsurface irrigation systems are also used, whose pipes are inserted in the soil. The benefit in reducing water evaporation is ensured by micro-irrigation, in which only small parts of the soil are irrigated.

So-called drippers are used, which are positioned above or below ground, as close to the roots as possible.

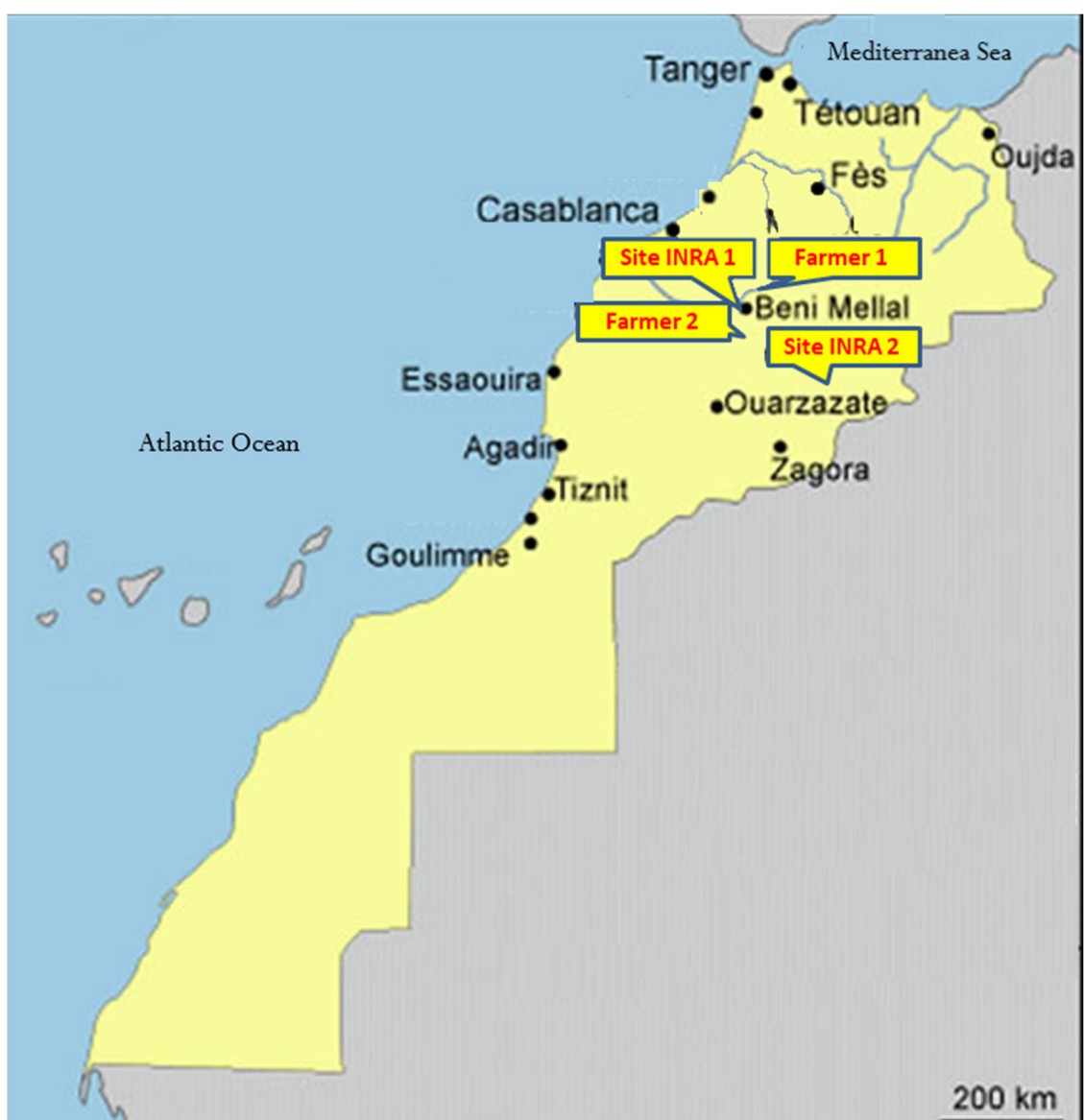

**Figure 5.** Distribution of the four experimental pilot sites for the installation of the SLECI irrigation system in Morocco [36].

### 3.3. Portugal

3.3.1. Nature Conditions

When trying to access and study irrigation problems, one must first understand what kind of climate characterizes the geographical area under analysis. In this case study, it must be considered a Mediterranean climate as illustrated in Figure 6, so available data about the water consumption in an apple culture already exists, and is presented in Figure 7, which shows the water consumption in liters per apple.

This kind of climate can be challenging when it comes to irrigation and water consumption, due to its low water availability, and climate changes [37], but mainly because of irregularity in the rain distribution through the seasons.

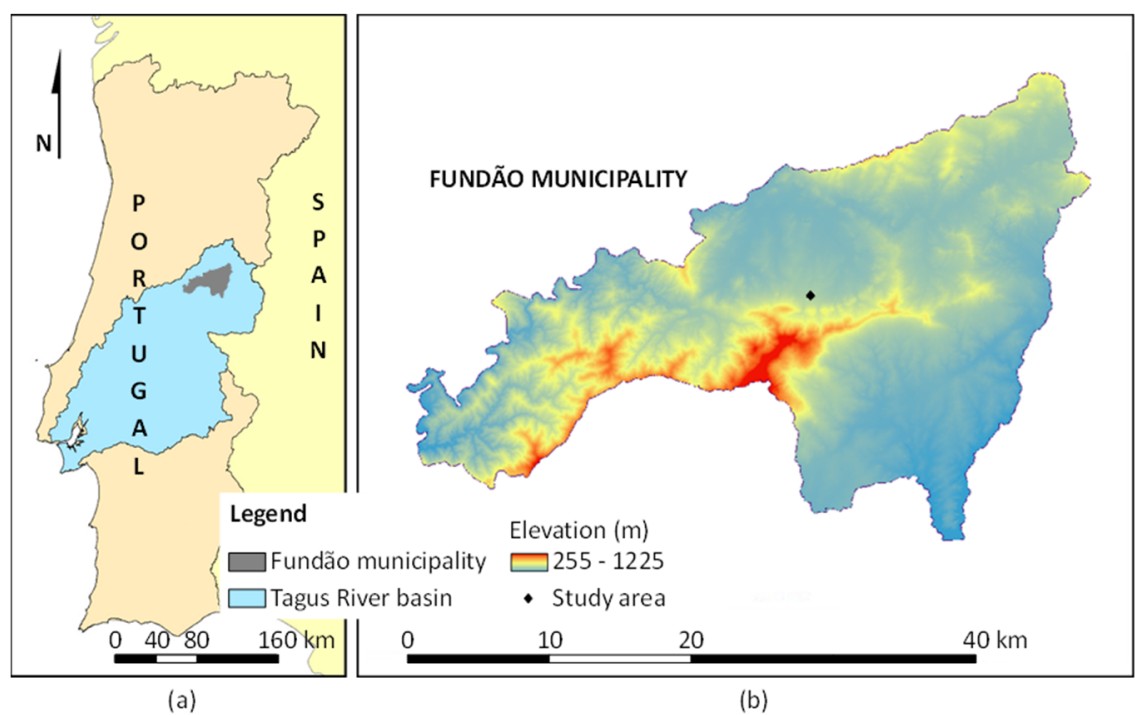

**Figure 6.** Location of study area (**a**) in Portugal, and (**b**) in Fundão (an author's figure).

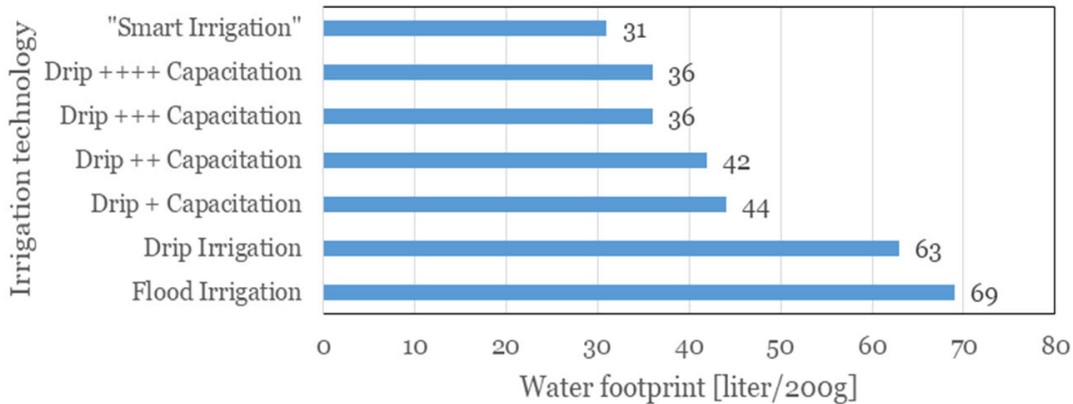

**Figure 7.** Effect of the application of technology for irrigation on the water footprint of an apple (Adapted from [37]).

### 3.3.2. Historical Evolution of Irrigation and Water Supply in Portugal

Portuguese current water regulation was highly marked by the 1919 approval of the first "Water Law" and the institutionalization of the plan for the Water Services [38,39]. Such initiatives were targeted at protecting water resources, both for the hydroelectric systems and for agricultural uses. In the 1940s, water legislation suffered a set of revisions with the concern of quality improvement [40].

Until the 1960s, irrigation was the largest consumer of water resources in Portugal. Concessions were subsequently made to hydroelectric companies, and hydroelectric production increased [41,42].

The 20th century brought a new focus to water policies and legislation, mainly focused on conveying water as an economic resource, creating technical infrastructures, and adjusting sanitary and environmental policies to water management [43].

The beginning of 2000 introduced the approval of the rivers' basin plans for the Portuguese international rivers, as well as for the national rivers. These were fundamental for spurring an integrated management of water resources and ecosystems.

One important and actual issue is the fact that Portugal and Spain share five common rivers, for which there were several bilateral agreements signed between the two governments, the most important of which being the ones targeted at regulating hydroelectric power generated on the international sections of shared river basins [42,44–46]. Of importance is the Albufeira Convention 2000 [47], which is fundamental for regulating common river basins relations and instruments, as it envisages mutual assistance and behaviours in extreme conditions, based on a sustainable water use of both countries.

In 2010, Portugal defined their strategy to adapt to climate change, with water scarcity being identified as one major concern. This Strategy reinforced the need to increase systems' resilience to such phenomena, and to compile a stock of knowledge on climate change impacts, so that it will be possible to design adaptation strategies to cope with extreme scenarios [42,48].

Conveying the following pressure on water scarcity, and in line with Bessa Santos et al. [49], Portugal receives a yearly average of about 700 mm of rain. However, the period from April to September is the drier one, notably in the southern and (central and northern), jeopardizing the season crops.

To cope with this, the Portuguese government established in 2018 a National Irrigation Program, which will be held until 2022. To renovate and update the current irrigation system, the initiative has been allocating 534 million euros for the creation of 90 thousand hectares of irrigated land, comprising new irrigation systems and the modernization and rehabilitation of existing ones.

The main aims of the program are: to boost sustainable agriculture with more integrated and efficient management of resources, namely water, soil, and energy; to respond to the impacts of climate changes on food security; and to implement a strategy that can strategically fight summer fires.

### 3.3.3. Main Irrigation Technologies in Portugal

The main irrigation technologies currently in use in Portuguese agricultural fields are conventional drip system, deficit irrigation, low-flow localized irrigation, underground irrigation [50], and the conventional sprinkler or center-pivot irrigation.

The conventional drip system has been a complete revolution when it comes to saving water, being able to save 28% to 35% on water losses [51]. The most used drippers are self-compensating, with flow rates around 2.2 L/h. The usual spacing between drippers is 0.75 m, but it varies depending on the kind of soil [50].

Due to the lack of water at some moments of the year, the producer always has the option to apply less water than advised on the crop during the whole year or in phases, always having in mind the life cycles of the plant that would minimize the effect on yield and quality of the production [52,53]. In this vein, Romero et al. [54] analyzed a set of water-saving irrigation strategies and methods targeted at increasing productive water use efficiency. The same goal was pursued by Cabral et al. [55], studying the effect of irrigation to improve crops' resilience from climate change and water scarcity. The authors concluded that rainfed plants experienced a negative effect on productivity, and that moderate water stress favorably affected fruit composition and yields. Regulated deficit irrigation takes into consideration the fact that plant production and quality responses to water stress vary throughout the phenological cycle. Partial root zone drying irrigation strategies entail, depending on the kind of soil, watering and drying around half of the root system of plants in cycles of 8–14 days [56].

The low-flow localized irrigation, or nano irrigation, is one of the best technologies when it comes to water retention in the soil. The drippers have a water debit of around 0.7 L/h [50], which is quite low value, and in that way, it becomes possible to irrigate a larger area, with a lower hydric and energetic cost. These technologies have also a low installation and maintenance cost, due to their low-pressure functioning.

Use of subsurface irrigation is still under investigation, but one of its great advantages is the plant being watered directly on the root, making it possible to reduce water losses

due to evaporation. On the other hand, this kind of technology has greater installation costs and is more likely to get damaged by rodent wildlife, in addition to which it is more likely that the system will clog due to salinity and sediments on the drippers [50]. There are nowadays many projects that investigate and use this kind of technology, the MED-WET project being the most relevant in this case study.

The sprinkler or center-pivot irrigation technology is quite useful when it comes to large scale irrigation of undulating areas. In general terms, it is an affordable technology. However, this kind of system can be easily erosive if the soil is not adequately protected. Having in mind that most Mediterranean agricultural fields are not regular, low infiltration rates are always a problem that must be considered. This kind of problem may result in significant run-off and erosion. As mentioned before, it is not an expensive technology but, in most cases, if not properly designed by an irrigation engineer, it can result in great water losses [57].

When it comes to new irrigation technologies in the Portuguese case study, we must consider two different projects. The proposed system is within the MED-WET project and the AQUA4D [36]. The AQUA4D system has a control unit that generates low frequency signals, and a processing unit that diffuses the generated resonance fields into the water. These resonance fields act directly on the physical structure of the water and modify the interactions between the liquid and solid interfaces. This allows a better penetration of water into the soil, a better infiltration into the soil micropores, and a greater retention leading to water savings. It also allows better dissolution, diffusion, and assimilation of mineral elements and fertilizers. MED-WET provides an innovative irrigation technology, ready to face the most serious problems in the Mediterranean region, namely the exponentially lower water availability through the years. This kind of subsurface irrigation technology must be taken into consideration in the future of water resource management, due to its low cost, low maintenance, and use of natural materials.

*3.4. Malta*

3.4.1. Nature Conditions

Situated in the center of the Mediterranean Sea, the Maltese Archipelago consists of three main islands–Malta, Gozo and Comino. With over half a million inhabitants in a total area of 316 km$^2$, the islands are by far the most densely populated member state of the European Union [58]. Climatologically, the islands are semi-arid, with an annual mean air temperature of 18.6 °C (mean minimum of 14.9 °C to mean maximum of 22.3 °C), and an annual total precipitation of 553 mm. Similar to other Mediterranean areas, the islands are subject to changing climatic patterns, with recent studies confirming increases in air temperature at significant rates (average annual air temperature anomaly increasing by +0.17 °C per decade), less frequent precipitation (decreased decadal amounts by −6 mm in the annual total precipitation), and a reduction in annual cloud cover by −0.05 Oktas [59,60].

With their dry climate and limited land cover, the islands lack any exploitable surface water resources such as rivers and lakes, making groundwater the only natural available freshwater resource. Geologically, the islands are characterized by two porous limestone formations, the Upper and Lower Coralline/Globigerina Limestones (separated by a less permeable formation of Blue Clay), each of which sustain respective bodies of groundwater. Being the only natural source of freshwater, groundwater has been a highly exploited resource over the years, with long term annual average abstraction from groundwater resources estimated to reach around 40 million m$^3$, and Water Exploitation Index (WEI+) values of 78%, reflecting the water-stressed nature of the Maltese aquifer system, as WEI+ levels over 40% are considered to reflect severe water stress [61]. Further, lower-lying groundwater is vulnerable to sea-water intrusion in response to abstraction activities, with chloride levels of up to 2000 mg/L [62].

For the purposes of the MED-WET project, the SLECI technology will be implemented in three field testing sites in the island of Gozo, and used for the irrigation of citrus groves,

grape vines, and annual crops. The pilot sites are located in Xaghra, Gharb, and Xewkija, as shown in Figure 8.

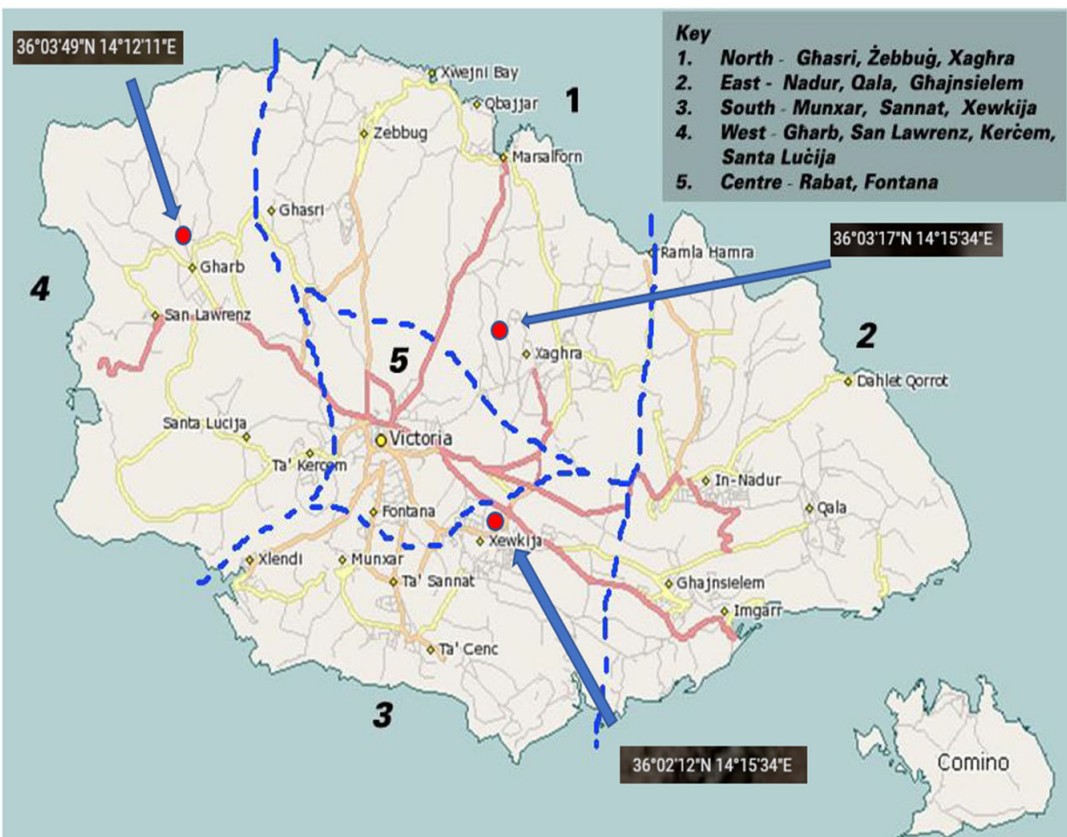

**Figure 8.** Location of fields at the 3 different pilot sites–Gharb, Xaghra & Xewkija.

The soils in Gozo are classified as (i) the Xerorendzina, locally known as "bajjad", (ii) the Carbonate raw soil, which includes the Fiddien Series and San Lawrenz Series, locally known as "tafli", and (iii) the Terra Rossa soil, which includes the Xaghra Series and Tas-Sigra Series, locally known as "hamri" (Figure 9). The Xerorendzina is the predominant soil on the island, while the Fiddien and San Lawrenz Series cover the slopes of the characteristic hilly terrain. The Terra Rossa is the most fertile but is quite shallow, only 30–40 cm deep [63]. Soils are largely influenced by the calcareous nature of the local geology, with high levels of calcium and magnesium carbonates commonly found throughout the entire soil profile.

### 3.4.2. Historical Evolution of Irrigation and Water Supply

In the past, agricultural practices were heavily dependent on annual rainfall, resulting in poor yields when precipitation was scarce. Farmers harvested rainwater in open reservoirs or wells hewn in the rock. Additionally, they exploited perched aquifers by digging horizontal galleries in the greensand rock just above the clay substrate. These galleries provided enough water to last through the dry season, though the volumes of water extracted varied from location-to-location and even tunnel-to-tunnel. The surrounding terraced fields just below the outlet of the water source were irrigated by gravity, using stone canals from terrace-to-terrace, and deep farrows within the field [64].

Since the Neolithic Era, cisterns were dug into the rock to provide a fresh water supply all year round. These can still be found near the Mnajdra temples on the southern coast of Malta. In medieval times, the Arabs introduced the stone canal flood irrigation system and built large rubble walls or reed canopies around orchards, to protect against the prevailing northwest winds as well as to reduce the water loss through evapotranspiration of the

plants. The Arabs were also the first to introduce the "Sienja"–a mechanical wooden structure powered by draft animals, to lift water for irrigation. The knights of St John exploited the perched aquifers by constructing aqueducts that carried spring water to the cities. They also dug large reservoirs under public squares and gardens, to harvest rainwater for domestic use as well as crop irrigation [64].

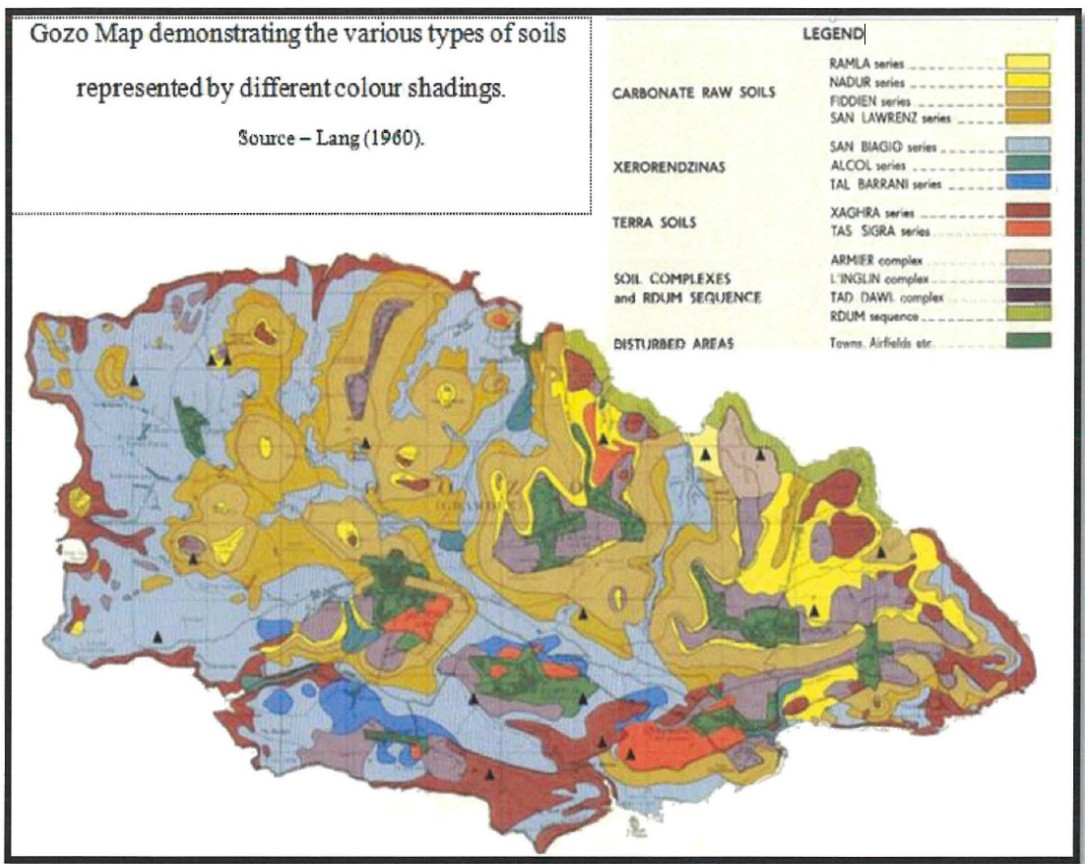

**Figure 9.** The map above illustrates the various soil types present in the Island of Gozo (Modified after Lang 1960 [62]).

In the last couple of centuries, the demand for water has exponentially increased, and the main sources of ground water are underground galleries dug down to the sea water level to collect the fresh water from the main aquifer. Additionally, there are several boreholes around the islands. Another seasonal source of fresh water for irrigation was and remains the valley dams, which retain the runoff water and eroded soil, to reduce permanent loss of these two resources into the sea [64].

### 3.4.3. Main Irrigation Technologies

The ever-increasing demand for fresh water for agriculture, industry, and the domestic and tourism sectors has led to investment in reverse osmosis plants over the past 50 years. There are currently four reverse osmosis plants in the Maltese islands, three in Malta and one in Gozo, and together they produce half the country's freshwater requirements [61].

Following the establishment of the reverse osmosis plants, investment in wastewater treatment plants soon began. The wastewater treatment plant effluent, also called "new water", is utilized by the farming sector for irrigation. In recent years, the local Water Services Corporation, through its new water program, has invested in the development of facilities for the further refinement of treated wastewater for use in the agricultural and commercial sectors, instead of groundwater resources. A high-end technological process has been developed that enables reclaimed water from the urban wastewater treatment

plants to pass through an additional three-barrier process – namely ultrafiltration, low-pressure reverse osmosis, and advanced oxidation. The availability of an all-year-round supply of good quality irrigation water increases security of supply to the agricultural sector, thereby increasing the crop production capacity and hence further contributing to food security. Malta's new water program aims to produce an annual volume of 7 million m$^3$ of reclaimed water, intended to partially address the water demand of the agricultural, landscaping, and industrial sectors [61].

Further, incentive and support programs have helped the agriculture sector to invest in highly efficient water irrigation technology. In fact, local farmers have shifted from surface irrigation (such as furrow and basin irrigation) to drip irrigation, where water is applied in droplets in a localized area [65]. Drip-irrigation technologies are currently applied to around 90% of all irrigated land. Support mechanisms such as financial and technical assistance programs are also in place to incentivize the introduction of new technologies, such as advanced irrigation systems. Efforts are also ongoing to enable improved management of groundwater use by the agricultural sector through the progressive metering of groundwater abstraction sources. Remote monitoring frameworks that allow comparative analysis of water use and agricultural land-use are also being developed to identify and address inefficient water use by the sector. Additionally, analytical methods for the estimation of irrigation water demand by use of satellite imagery are currently being developed through the WARM-EO Project, a joint research initiative between the local Energy and Water Agency and the University of Malta (Msida, Malta), which will allow the identification of inefficient water users, who can then be supported to better manage their water use [61].

## 4. Discussion

The results of the present research contribute to the outline of the three key technologies for irrigation and water supply for small-holder farmers in the Mediterranean region:

1.  SLECI (Self-Regulating Low Energy Claytube Irrigation);
2.  Desalination technologies;
3.  Engineering constructed wetlands.

### 4.1. SLECI

The SLECI irrigation technique is an innovative buried clay irrigation system that allows the exploitation of unconventional water resources at low cost. The SLECI irrigation system is based on a natural physical principle in the use of ceramics as a porous, water-permeable material, and belongs to the micro-irrigation systems. The SLECI emitter is made with natural clay materials and available locally. In this technology, cylindrical clay tubing bodies (Figure 10) are connected to each other with thin tubes and buried in the soil near the plant root. The hose system is then connected to the water reservoir (e.g., poly tank, river, lake, water pressure connection) via connectors. The water inflow typically takes place without additional energy (e.g., pump system). During irrigation, the inner space of the clay body is first filled with water, and a so-called suction effect is created on the water in the hoses. The clay material consists of a multitude of micropores that function together as a network and cause capillary forces. These are in turn dependent on:

1.  The surface tension of the fluid medium;
2.  The adhesion at the interface between liquid and solid;
3.  The size and shape of the pores;
4.  The interactions of the solid with the polar water.

The surrounding soil works according to the same principle. This also consists of a multitude of pores, which in turn exert a suction effect on the clay body. A suction pressure on the clay body caused by osmosis becomes greater, the lower the soil moisture becomes. As soon as this is greater than the water retention force of the pores of the clay body, the

clay body releases the water to the soil. Accordingly, the water continues to flow in the hose system, as the clay body in turn exerts a suction pressure on the water reservoir.

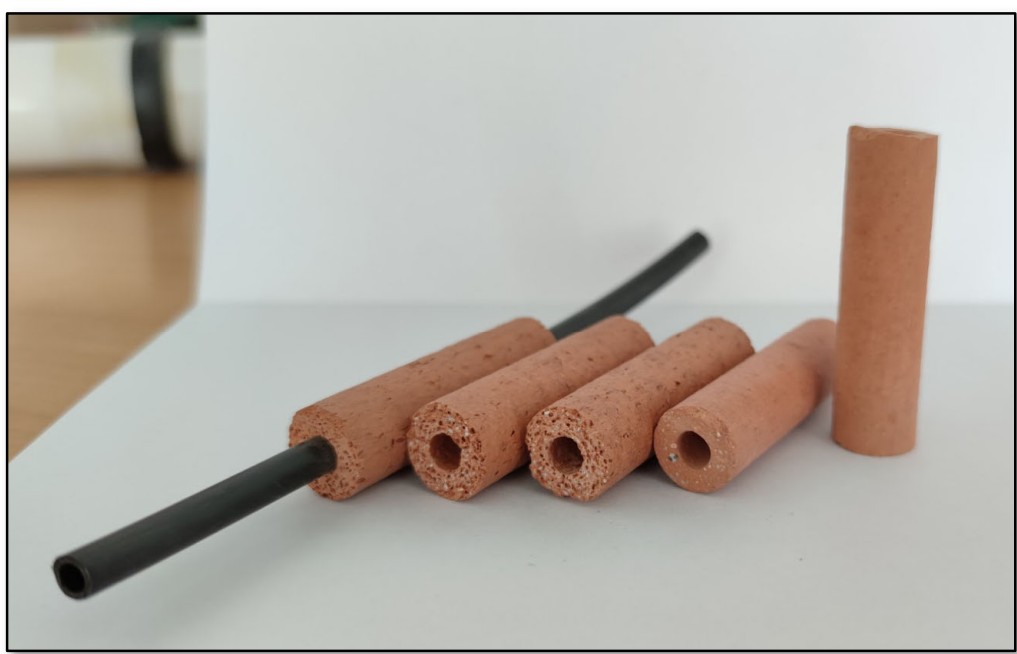

**Figure 10.** Clay-bodies of the SLECI–irrigation system (an author's contribution).

Sustainability and environmental awareness play a decisive role in the evaluation of this technology. The SLECI irrigation system is built on the principle of the natural suction effect, which is ensured by energy-intensive pump systems, among other things. However, a certain pre-pressure on the SLECI irrigation system is required to maintain tightness against air infiltration. This means that the water source must be slightly above the SLECI pipes. In the event that water tanks are used, they will need to be refilled periodically once the water level has dropped. This energy expenditure can be minimized if the water source is geographically located on higher terrain, for example in the form of a basin. An exact assessment therefore depends on the local environmental conditions of the usable area.

The irrigation system works with a low hydrostatic pressure, so that only simple pressure regulators must be installed. In this case, the installation level of the system is slightly below the water level of the reservoir. The hoses as well as most of the connectors are made of polyethylene throughout. This type of polyolefin is particularly recyclable, especially as this plastic is not used in composite materials, which are very difficult to break down into their individual components. Polyethylene can be remelted as a thermoplastic or recycled to make new products. The clay material consists of naturally occurring solids. In terms of water consumption, the way the porous clay bodies work results in enormous water savings. With conventional irrigation technologies, much of the water is lost through unused water seepage and evaporation. Subsurface irrigation impedes evaporation at the soil surface. The physical interactions of the surrounding soil with the clay bodies, reduces water seepage into deeper soil layers due to gravity.

On the other hand, the following points should also be viewed critically.

The energy required to produce the burned clay bodies: among other things, this involves the use of powerful furnaces that are typically also used in pottery.

Additionally, in all underfloor systems, the materials must be dug up again when they are no longer used or have exceeded their service life. Leaving the hose lines in the ground must be avoided in any case.

The use of sealing materials, which are used in most irrigation technologies, cannot be avoided with this technology either. These should be used in small quantities, if possible, as they can have a negative impact on the environment.

Therefore, the SLECI technology, as compared to conventional irrigation technologies, has higher initial costs and requires training in its effective use. In addition, for the underground irrigation lines, costly excavations or drilling to root depth must be carried out. On the other hand, there are the following advantages:

1.  Water conservation: especially in arid or dry areas with water shortages, this can be a decisive factor for the economic and ecological development of the country;
2.  Practical plug-in system based on the "modular principle": depending on field conditions, the irrigation system can be adapted;
3.  Recyclability: irrigation components can be easily disassembled and recycled after use;
4.  High efficiency: plant development is promoted by the principle of the naturally occurring suction effect, which can have a positive effect on the crop yield.

The economic perspective of SLECI is considered by its (i) implementation, and (ii) economic efficiency. As to (i) implementation, the SLECI irrigation system will be used to irrigate a wide range of crops found in agriculture. As part of the MED-WET project, tests will take place on test fields in Morocco and Portugal. The system will be made available in particular to smallholder farmers who have predominantly worked with conventional technologies.

Although it is difficult to transfer knowledge about this new irrigation technology to the farmer, as he tends to trust the old established technology, the effects of climate change force him to rethink. Having less water available for crop irrigation can result in a livelihood issue, as the farmer depends on the yield of his crop. In many areas, the loss of a large part of the harvest already means the financial end for a small farmer, as he hardly has any financial reserves to compensate for the crop failure. A risk-benefit assessment is basically carried out by each smallholder farmer. To establish new technologies, persuasive efforts such as on-site demonstrations, the transfer of experience through third parties, and the public dissemination of successes through the use of this technology, can be made, among other things.

Water scarcity is not only a consequence of climate change, but is also due to excessive water consumption, which is caused by the fact that the groundwater level has dropped because more water has been taken from springs than could be replenished by precipitation.

The demand for water-saving irrigation technologies is increasing and has meanwhile also reached the interest of politics. An important criterion is the crop yield that can be achieved by the irrigation technology.

The SLECI irrigation technology can be adapted to the respective needs. Plants with higher consumption can be watered as well as plants with low consumption. The individual components can be plugged together via connectors and used over an entire irrigation field, depending on requirements. However, it is essential to observe the boundary conditions of the system and to study the instructions carefully.

Regarding (ii) economic efficiency, the initial cost of the SLECI irrigation system is in any case higher than that of simple irrigation pipes that only use boreholes as emitters for water delivery. This is simply because this technology consists of several components that have to be assembled. In addition, bore-holes or trenches must be laid so that subsurface irrigation can be applied. However, this is offset by the cost of water consumption, which is far lower than conventional technologies for subsurface irrigation and especially for this technology.

A productive development of plants depends to a large extent on the moisture of the surrounding soil. It is important that neither the permanent wilting point (irreversible wilting) is exceeded nor the field capacity point (water content that can be held in the soil against gravity) of the soil far from groundwater is undershot. The moisture content must thus be kept between these two values and water stress must be avoided. It is important that neither the permanent wilting point (irreversible wilting) is exceeded nor the lowest value of the soil's field capacity (water content that can be held in the soil against gravity) is fallen below. Soil type, pore distribution, soil structure and the chemical-physical interactions in the soil are important factors here. The working principle of porous water reservoirs in the

soil leads to plants getting the water they need at that moment. The usable water is always made available to the root through the water transport in the soil, without the plant having to restrict its growth. So-called waterlogging is also prevented, as the suction effect on the clay bodies is decisively reduced with increasing soil moisture.

The water savings of the clay body irrigation system refers to the amount of water saved that is needed to grow the crops compared to alternative irrigation systems that are delivered through the outlets of the irrigation system. For a comparison, the crop area, the number of plants, the development stage of the plants and the local conditions such as soil properties and climatic conditions must be known and similar. A high water saving on the same crop area compared to the conventional technology used can have a positive impact on the environment in sustainable agricultural use. The decreasing groundwater level due to climate change could be accelerated less if the water extraction for the crops is done from groundwater. This can facilitate the supply of drinking water to residents located along the fields. If the groundwater level is too low, the supply can only be provided by transport, which continues to increase the price of water.

Financial savings and decreasing marginal costs are often observed in irrigation projects. Both create problems for funding systems large enough to achieve economies of scale. Before a positive investment decision in favor of the SLECI technique, it seems necessary to justify in terms of financial and economic profitability, as well as in terms of efficiency and productivity in selected Mediterranean countries. This is achieved by understanding the determinants of the costs, benefits, efficiency, and productivity parameters of water demand for irrigation crops, using the innovative SLECI irrigation technique compared to alternative irrigation.

### 4.2. Desalination Technology

Approximately 97.5% of the available water on the planet is saline [66]. The remaining 2.5% fresh water can be used for irrigation as well as for human consumption. The scarcity in water resource is more accentuated for small islands with no permanent water body. Farming community has always addressed this water scarcity problem through development and adaptation of techniques to maximize water harvesting and efficient distribution. One aspect of these technologies is to produce fresh water from saline sources using desalination technologies, namely Reverse Osmosis (RO) technology and evaporation/condensation technologies within a greenhouse.

The Solar Desalination Greenhouse (SDG) will include a series of technologies, namely reverse osmosis, dehumidifiers, and halophytes plants, that collectively will produce fresh water from a saline source. An SDG with bio-based solutions has high potential in meeting the needs of modern agricultural practices with low impact on environment yet high profitability for farmers. By means of active and passive condensation, the SDG produces fresh water. The method has several advantages, such as flexibility in capacity, moderate installation and operating costs, simplicity, possibility of using low temperature, and the use of renewable energy (e.g., solar, geothermal, recovered energy or cogeneration) [67].

### 4.3. Engineering Constructed Wetlands

The Egyptian wastewater treatment reuse code was developed by the Egyptian Ministry of Housing, Utilities and Urban Communities (MHUUC) and adopted in 2015 [25]. For rural smallholder farmers and remote communities comes the low-cost technology of wetland treatment as a suitable alternative that does not need energy or sophisticated equipment to operate, yet provides the smallholder farmers with sufficient treated water suitable for irrigating their local crops and trees while maintaining the human health and environmental safeguards according to the Egyptian code (Figure 11). For every 5 households (25 persons in average) in rural communities, an engineering constructed instream wetland site of 100 m $\times$ 25 m (two cells) could be enough to treat their domestic and agricultural drainage waters, resulting in a daily treated wastewater of about 40 m$^3$/day to irrigate their local crops and trees. The cost of constructing and operating such a nature-based site

will be affordable by the rural communities because its components are from the natural environment, with no electrical or mechanical parts at all.

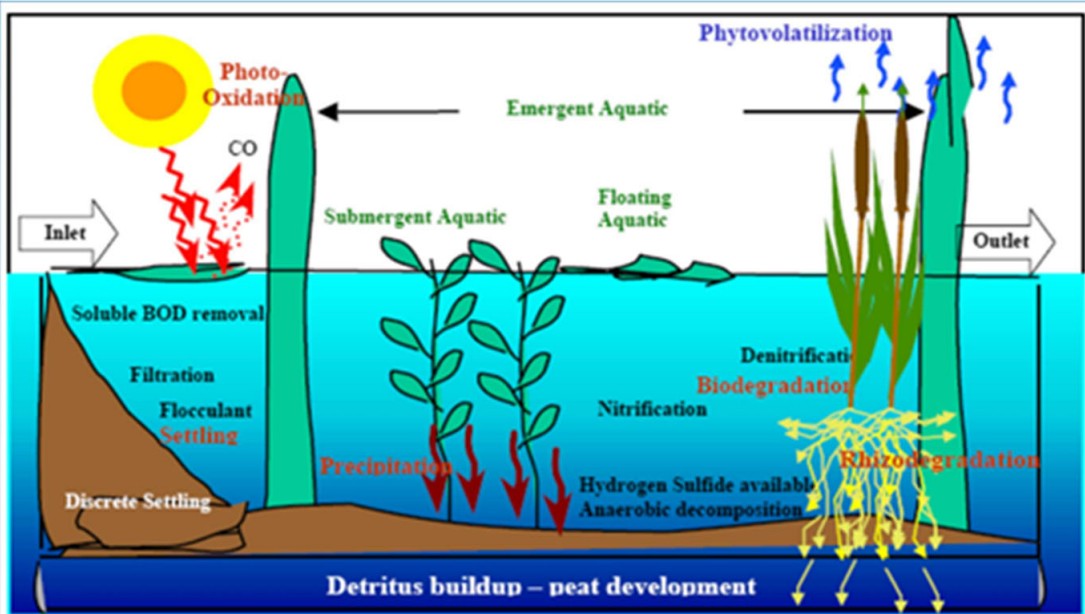

**Figure 11.** The physical layout of an engineering constructed wetland, as a cheap, low energy and efficient technology for wastewater treatment in remote communities [67].

The following points describe the engineering constructed instream wetland technology used for irrigation for smallholder farmers located in the desert land called Sekem for Land Reclamation farm in El-Wahat El-Baharia, Egypt:

- Engineering constructed instream wetlands are longitudinal cells with 50 m length (trapezoidal cross-section of 1.5 m depth, 1:3 side slope and 1.0 m bottom width). Flow of wastewater has shallow depth of 60 cm (hydraulic depth) with low-speed waters;
- Provided with gravel and sand filters, sharp crested weirs, and substrate medium to support rooted and floating vegetation. It consists of plants, biofilms, soil, micro-organisms, and organic materials, to naturally treat and remove the water pollution;
- The hydraulic retention time in the treatment cells is estimated to be 3 days, with treated effluent of about 20 $m^3$/day per cell.

In such engineering constructed instream wetlands, the aerobic and anaerobic reactions occur without energy (cheap technology) as well as sedimentation, filtration, and plants abstraction, to efficiently treat almost all pollutants (removal of biological load, fecal coliform, *E. coli* bacteria, pathogens, complex nutrients and phosphorus compounds, and metals).

The advantages of such engineering constructed instream wetland technology system are as follows:

- Efficient and cheap technology for treating domestic wastewater and agricultural drainage water;
- Increase the available water resource (non-conventional water) in scarce water regions;
- Reclaims nutrient-rich effluent for irrigation purposes;
- Preserves the groundwater and surface ponds from pollution, leading to reduction of the environmental impacts;
- Useful for safe sludge management and on-site reuse with zero waste;
- Good application of the circular economy concept for the smallholder farmers;
- Supports the local business creation and smallholders' irrigation in remote com-munities.

## 5. Conclusions

This study explores potential key technologies for irrigation and unconventional water resources for smallholder farmers in the selected Mediterranean countries (Egypt, Malta, Morocco, and Portugal). The literature survey allows the conclusion that, while Mediterranean countries share many common features in terms of climate, water and land resources, and development issues [18], the selected countries in the Mediterranean region (Egypt, Malta, Morocco, and Portugal) differ in terms of type of crops, water management regulations, labor force availability, financial sustainability, and economic perspective. This contrasting picture, that the Mediterranean countries are simultaneously similar and different, demands an individual approach when it comes to deciding what kind of key technology for irrigation and water supply to use in a particular country. The trade-off between the benefits and costs associated with the use of a technology for irrigation and water supply must be considered for in order to ensure the sustainable development of smallholder farmers in the Mediterranean region. In this same line of reasoning, innovative irrigation systems that allow the exploitation of unconventional, largely unused water resources must be developed. Thus, low-cost solutions with natural and locally available materials (low technology, low energy, low cost, easy to use) have been proposed.

Despite some limitations, such as the engagement of only four Mediterranean countries, and the reduced number of technologies for irrigation and water supply for smallholder farmers in the Mediterranean region, the results could be used in the same conditions, by the Mediterranean countries.

The set of exploratory results presented here provide an important set of implications, not only for policymakers, but also for higher education institutions and research structures, and for small farmers. Firstly, policymakers need to launch specific financing lines and incentives for fostering smart technology transfer mechanisms targeted to tackle water scarcity and climate change mitigation. Secondly, higher education institutions and research structures should be incentivized to be engaged in open innovation platforms, integrating local government, research institutions, and small farmers. Thirdly, the small farmers need to have special access to financed low-cost smart irrigation technologies, in order to reduce the adoption time of these types of technologies.

In terms of future work, the research consortia are targeted to examine the three key technologies (i.e., SLECI, desalination technology, and engineering constructed wetlands) in the four Mediterranean countries and to increase the still-limited knowledge about the benefits associated with the adoption of those technologies for irrigation and water supply of smallholder farmers in the region. In addition, comparative analyses on the three key technologies (SLECI, desalination technology, and engineering constructed wetlands) in Egypt, Malta, Morocco, and Portugal will be carried out, embracing financial, economic, efficiency, and productivity perspectives, and applied to the three key technologies oriented to sustainable development.

**Author Contributions:** Conceptualization, D.P., J.C.C.L., P.D.G., C.F., W.K., N.W., H.E.Y., B.B., I.F., J.S., H.H. and J.Z.; methodology, J.Z. and N.W.; validation, D.P., J.C.C.L., P.D.G., C.F., W.K., N.W., H.E.Y., B.B., I.F., J.S., H.H., S.C., F.B., M.B., J.M., R.M. and J.C.; formal analysis, D.P., J.C.C.L., P.D.G., C.F., W.K., N.W., H.E.Y., B.B., I.F., J.S., H.H., J.Z., S.C., F.B., M.B., J.M., R.M. and J.C.; investigation, D.P., J.C.C.L., P.D.G., C.F., W.K., N.W., H.E.Y., B.B., I.F., J.S., H.H., J.Z., S.C., F.B., M.B., J.M., R.M. and J.C.; data curation, D.P., J.C.C.L., P.D.G., C.F., W.K., N.W., H.E.Y., B.B., I.F., J.S., H.H., J.Z., S.C., F.B., M.B., J.M., R.M. and J.C.; writing—original draft preparation, D.P., J.C.C.L., P.D.G., C.F., W.K., N.W., H.E.Y., B.B., I.F., J.S., H.H., J.Z., S.C., F.B., M.B., J.M., R.M. and J.C.; writing—review and editing, D.P., J.C.C.L., P.D.G., C.F., W.K., N.W., H.E.Y., B.B., I.F., J.S., H.H., J.Z., S.C., F.B., M.B., J.M., R.M. and J.C.; visualization, D.P., J.C.C.L., P.D.G., C.F., W.K., N.W., H.E.Y., B.B., I.F., J.S., H.H., J.Z., S.C., F.B., M.B., J.M., R.M. and J.C.; supervision, D.P., J.C.C.L., P.D.G., C.F., W.K., N.W., H.E.Y., B.B., I.F., J.S., H.H. and J.Z.; project administration, D.P., J.C.C.L., W.K., N.W., B.B., H.H., J.Z., M.B., J.M., J.C.; funding acquisition, D.P., J.C.C.L., P.D.G., C.F.,W.K., N.W., H.E.Y., B.B., I.F., J.S., H.H., J.Z., S.C., F.B., M.B., J.M., R.M. and J.C. All authors have read and agreed to the published version of the manuscript.

**Funding:** This research carried out within the project MED-WET "Improving MEDiterranean irrigation and Water supply for smallholder farmers by providing Efficient, low-cost and nature-based Technologies and practices" (Project ID 1646) funded by Partnership for Research and Innovation in the Mediterranean Area (PRIMA) program.

**Institutional Review Board Statement:** Not applicable.

**Informed Consent Statement:** Not applicable.

**Data Availability Statement:** Data is contained within the article.

**Conflicts of Interest:** The authors declare no conflict of interest. The funders had no role in the design of the study; in the collection, analyses, or interpretation of data; in the writing of the manuscript; or in the decision to publish the results.

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
