# Peer review of "Exploring Irrigation and Water Supply Technologies for Smallholder Farmers in the Mediterranean Region"

_sustainability, doi:10.3390/su15086875_

Round 1

Reviewer 1 Report

Taking Egypt, Morocco and Portugal as the representative countries in the Mediterranean region, the author described in detail the water resource conditions, irrigation and water supply, irrigation wastewater and other aspects through literature review, field survey and observation, and introduced new seawater desalination technology, engineering constructed wetlands, self-regulated low-energy clay pipe irrigation (SLECI) and other technologies. The manuscript has done a lot of work, but it still needs to be modified from the following aspects.

1. Please add why Egypt, Morocco and Portugal are selected as the representative countries in the Mediterranean region.

2. It is recommended to redraw picture 3. The resolution of pictures 4 and 7 is not high. It is recommended to improve the resolution of pictures.

3. The second part is about materials and methods, which are various in content and recommended to be simplified.

4. The description items of the three countries are not uniform, so it is recommended to unify their description objects.

Author Response

Dear Reviewer,

The file with the updated paper is attached.

The changes in the paper are highlighted by the blue colour.

Here are our answers to Reviewer 1:

Reviewer comment

Answer to the reviewer

Paper Page with the relevant updates

Line

English language and style are fine/minor spell check required  

English has been re-checked

1-27

1. Please add why Egypt, Morocco and Portugal are selected as the representative countries in the Mediterranean region.

Under Med-Wet project, a set of field testing sites will be object of study, in mediterranean countries, such as Egypt, Malta, Morocco and Portugal. For the purpose of the present article,Egypt, Morocco and Portugal, were selected due to the status of the study which is further developed and due to the fact that the three countries are located in different levels of arid regions where climate change is already severely affecting agricultural production, also in diverse stages. As so, the selected sites are representative of all pilot sites, involving farmers in participatory research, co-development, capacity-building and demonstration activities.

Page 2

53-54

2. It is recommended to redraw picture 3. The resolution of pictures 4 and 7 is not high. It is recommended to improve the resolution of pictures.

updated

Figure 3 now is Figure 1 on Page 4

Figure 4 now is Figure 5 on Page 12

Figure 7 is now Figure 11 on Page 22

120

445

859

3. The second part is about materials and methods, which are various in content and recommended to be simplified. 

Simplified, updated and re-structured

Pages 5-7

166-222

4. The description items of the three countries are not uniform, so it is recommended to unify their description objects. 

We have uniformized the information for each country as follows:

3.1. Country name

3.1.1. Nature conditions

3.1.2 Historical evolution of irrigation and water supply

3.1.3. Main irrigation technologies

Pages 7-17

229-652

Looking forward to receiving your further comments.

Best regards,

Jelena

Reviewer 2 Report

Thank you for the chance to review your paper. The topic(s) are relevant to the future development of sustainable irrigation. However, as currently presented, the paper is not ready for review and requires reorganization to enable the reader to understand your work, arguments and findings. I found it hard to understand what stage the project/studies are at - it would be useful to provide some information on the timeline of the MED-WET and AQUA4D efforts. 

I was disappointed that the paper introduced the ideas about cost-benefit and economic analysis, but then presented no numbers about the costs and/or benefits of the technologies discussed.

I have made some suggestions on terminology within the annotated file and recommend the team of authors consider how to address the suggestions for improvement. Before resubmitting for further consideration, I suggest a smaller team of authors (perhaps one or two leading authors) review the revised paper to ensure consistency of style and level of information presented.

With careful editing the paper could be significantly shorter without loosing necessary details. You may wish to consider splitting the paper into multiple papers - one the literature review and introduction to the study and sites; a second on the findings from the field sites and recommendations for adoption/dissemination of the findings.

Author Response

Dear Reviewer,

The file with the updated paper is attached.

The changes in the paper are highlighted by the blue colour.

Here are our answers to Reviewer 2:

Reviewer comment

Answer to the reviewer

Paper Page with the relevant updates

Extra comments

1. Extensive editing of English language and style required  

English has been re-checked for several times

1-27

2. The paper is not ready for review and requires reorganization to enable the reader to understand your work, arguments and findings.

The paper has been re-organised: the title was changed, some parts are deleted, each country description was uniformized

Pages 1-27

3.I found it hard to understand what stage the project/studies are at - it would be useful to provide some information on the timeline of the MED-WET and AQUA4D efforts. 

The information about the MED-WET and AQUA4D efforts was extended: The AQUA4D system has control unit that generates low frequency signals and a pro-cessing unit that diffuse the generated resonance fields into the water. These resonance field act directly on the physical structure of the water and modifies the interactions between the liquid and solid interfaces. This allows a better penetration of water into the soil, a better infiltration into the soil micropores, a greater retention leading to water savings. It also allows better dissolution, diffusion and assimilation of mineral elements and fertilisers.”

MED-WET: Pages 2-3

AQUAD4D: Page 14

54-81

535-548

4. I was disappointed that the paper introduced the ideas about cost-benefit and economic analysis, but then presented no numbers about the costs and/or benefits of the technologies discussed.

These parts were deleted as the project has not reached the level of data collection yet

4. I have made some suggestions on terminology within the annotated file and recommend the team of authors consider how to address the suggestions for improvement. Before resubmitting for further consideration, I suggest a smaller team of authors (perhaps one or two leading authors) review the revised paper to ensure consistency of style and level of information presented.

Thank you for your comments!

We have re-checked the whole paper. 3 co-authors did the final editing.

With careful editing the paper could be significantly shorter without loosing necessary details. You may wish to consider splitting the paper into multiple papers - one the literature review and introduction to the study and sites; a second on the findings from the field sites and recommendations for adoption/dissemination of the findings.

Thank you!

We deleted some parts of the paper which referred to the later developments of the project. The focus of the remaining text is on the introduction of each country Nature conditions, its Historical evolution of irrigation and water supply, and each country’s

Main irrigation technologies

Looking forward to receiving your further comments.

Best regards,

Jelena

Round 2

Reviewer 2 Report

Thank you for the efforts to improve the first reviewed version of the paper. The paper is considerably better, but still falls short of the standard I would expect for submission to a leading journal. 

I note you have taken on board many of the suggestions from the reviewers; however I also note their remain a number of questions and requests for clarification in the comments that remain unanswered. Please address these before submitting for a further review.

I appreciate the efforts to have a smaller group of authors lead finalizing the writing of the draft paper - however before submitting a further review draft, please take the time to address the formatting and other issues that remain in this paper. There are multiple fonts, paragraph styles, etc that detract from the review process. Most reviewers are busy and fixing your presentation issues is not a high priority.

Author Response

A lot of thanks for the review!

Here are the answers to the reviewers' comments:

Reviewer comment

Answer to the reviewer

Paper Page with the relevant updates

Line

English language and style are fine/minor spell check required  

English native speaker has re-checked the whole paper

Is the content succinctly described and contextualized with respect to previous and present theoretical background and empirical research (if applicable) on the topic?

The text was re-checked

Are all the cited references relevant to the research?

As it was not a must, they remained unchanged

Are the research design, questions, hypotheses and methods clearly stated?

They were re-checked

Are the arguments and discussion of findings coherent, balanced and compelling?

the first 3 paragraphs in the discussion as well as figure 9, as these focus on Morocco, rather than providing an overall discussion are mover to Section 3.3.3. Morocco Main irrigation technologies.

Pages 13-14

507-536

For empirical research, are the results clearly presented?

They were re-checked and updated

Are the conclusions thoroughly supported by the results presented in the article or referenced in secondary literature?

They were re-checked and updated

formatting

Updated

Round 3

Reviewer 2 Report

Thank you for your positive responses to the earlier reviews and suggestions. I find the paper considerably improved compared to the previous submissions.

I find the paper is somewhat long for the material presented and would recommend the authors consider whether some sections could be streamlined and still convey the major issues and thoughts of the team. I do recognize that with a large number of authors, four study countries, and three areas of technology arriving at a succinct summary is challenging. 

I have made a few suggestions or recommendations in the attached file for the teams consideration. Overall I think the paper would be best presented as a set of related papers (i) an overview of the project, sites, and technologies; (ii) a paper based on each country or each technology; and (iii) at a later date a paper summarizing the findings and recommendations from the study.

Author Response

  1. Figure 5 was updated.
  2. The paragraphs indicated by the review in the PDF file were re-considered and corrected.
  3. In regard to the reviewer's comments to publish a set of related papers, our concern is to highlight the commonalities between the countries in the Mediterranean region for the introduction of the 3 irrigation technologies (SLECI, desalination and Wetlands). After that, we intend to analyse the use of the 3 technologies for their adaptation to local conditions.